# Mast Cells and Vitamin D Status: A Clinical and Biological Link in the Onset of Allergy and Bone Diseases

**DOI:** 10.3390/biomedicines10081877

**Published:** 2022-08-03

**Authors:** Giuseppe Murdaca, Alessandro Allegra, Alessandro Tonacci, Caterina Musolino, Luisa Ricciardi, Sebastiano Gangemi

**Affiliations:** 1Department of Internal Medicine, Ospedale Policlinico San Martino, 16132 Genoa, Italy; 2Department of Human Pathology in Adulthood and Childhood “Gaetano Barresi”, Division of Hematology, University of Messina, 98125 Messina, Italy; cmusolino@unime.it; 3Clinical Physiology Institute, National Research Council of Italy (IFC-CNR), 56124 Pisa, Italy; atonacci@ifc.cnr.it; 4Department of Clinical and Experimental Medicine, School and Operative Unit of Allergy and Clinical Immunology, University of Messina, 98125 Messina, Italy; luisa.ricciardi@unime.it (L.R.); gangemis@unime.it (S.G.)

**Keywords:** mast cell, vitamin D, allergy, osteoporosis, mastocytosis, bone diseases, immune response

## Abstract

The immune system is made up by an extremely composite group of cells, whose regulated and harmonious activity is fundamental to maintain health. The mast cells are an essential effector of inflammatory response which is characterized by a massive release of mediators accumulated in cytoplasmic secretory granules. However, beyond the effects on immune response, mast cells can modify bone metabolism and are capable of intervening in the genesis of pathologies such as osteoporosis and osteopenia. Vitamin D is recognized to induce changes in bone metabolism, but it is also able to influence immune response, suppressing mast cell activation and IgE synthesis from B cells and increasing the number of dendritic cells and IL-10-generating regulatory T cells. Vitamin D deficit has been reported to worsen sensitization and allergic manifestations in several different experimental models. However, in clinical situations, contradictory findings have been described concerning the correlation between allergy and vitamin D deficit. The aim of this review was to analyze the close relationships between mast cells and vitamin D, which contribute, through the activation of different molecular or cellular activation pathways, to the determination of bone pathologies and the onset of allergic diseases.

## 1. Introduction

Allergies and osteoporosis are disorders with a great incidence in the overall population and constitute a real danger for community health [1]. Furthermore, several experimental findings have shown a close correlation between the two conditions. Many allergic conditions such as asthma, eczema, chronic respiratory pathologies, and pollen allergy are correlated with fracture risk [2,3], while increased percentages of bone diseases are described in patients with allergic syndromes, in both adults and children [4].

For instance, the connection of asthma with osteoporosis is well established, although up until the recent past, most studies attributed the correlation to the unfavorable effect exerted by protracted steroid therapies on the bone. New findings have instead highlighted the responsibility of the systemic inflammation present in allergic diseases in the osteoporosis onset [5,6], and a correlation between allergies and osteoporosis has been also demonstrated in patients suffering from chronic rhinosinusitis [5].

As for the mechanisms capable of connecting bone diseases and allergic pathologies, a correlation between bone and immune effectors is now recognized, and several relevant communications are seen in the bone milieu, such as the enrolment and growth of T cells [7]. It is also known that bone cells and immune effectors originate from the same precursors in the bone marrow (BM) and that the same compounds can control bone cell’s proliferation and activity, immune response, and haematopoiesis [8]. Moreover, the destiny of hematopoietic stem cells is regulated by autocrine and paracrine systems, these phenomena being controlled by bone tissue, osteoclasts, osteoblasts, and immune effectors interacting with each other. Several blood cells, including basophils, mast cells, eosinophils, lymphocytes, and neutrophils contribute to this composite crosstalk, which causes the allergic inflammation. This condition can generate cytokines, reactive oxygen species, chemokines, and lipids leading to a plethora of effects which are able to elicit bone modifications [9].

For instance, the Cysteinyl leukotrienes (CysLTs) are a group of lipid mediators originating from arachidonic acid [10]. The CysLTs receptor (CysLTR1) has a relevant effect in the onset of asthma, and montelukast is a CysLTR1 antagonist employed for the therapy of asthma. An experimentation demonstrated that montelukast effectively inhibits RANKL-caused osteoclast generation and bone damage in vivo [11].

However, one of the main actors of this communication is represented by mast cells (MCs), which play an extremely important role both in the onset of allergic pathologies and in diseases of bone metabolism, and the effect exerted by vitamin D on these cells could constitute a unifying element for apparently heterogeneous alterations (Figure 1).

The aim of this review was to detect the close connections between mast cells and vitamin D, which contribute, through the activation of different molecular or cellular activation pathways, to the determination of bone pathologies and the onset of allergic diseases.

## 2. General Considerations on Vitamin D

Vitamin D3 (cholecalciferol) is mainly produced in the epidermis when pre-vitamin D3, the compound originating from 7-dehydrocholesterol after ultraviolet-B (UVB) irradiation of the skin, undertakes thermal isomerization. The transformation of vitamin D3 to its active form 1α,25(OH)2D3, is due to a sequence of hydroxylation procedures, initially caused by liver cytochrome P450 proteins, which can produce the transitional metabolite, 25OHD3 and, subsequently, 25-hydroxyvitamin 1αhydroxylase in the renal tubule to generate 1α,25(OH)2D3. This compound operates by joining to the vitamin D receptor (VDR), which provokes the enrolment of the retinoid X receptor, to generate a heterodimeric structure that affects vitamin D (VD) response factors in the promoter areas of genes. According to the concurrent joining of nuclear co-stimulators or coinhibitory factors, the complex can operate as a ligand-dependent stimulator or an inhibitor of gene transcription [12,13,14].

VD deficit has been recognized as a main health problem, which is constantly more prevalent [15]. Even though there is no agreement on the ideal concentrations of 25(OH)D, a VD deficit is identified when concentrations of VD less than 50 nmol/L occurs [16].

All organs and cells feature a VDR, including immune and bone cells, skin, brain, gonads, and heart cells. A reduction on VD levels may change the activity of these tissues. This factor explains the extensive effects of VD and justifies why a decrease in VD levels has been related with several different chronic pathologies. Regarding the subject of our analysis, VD has a main action in calcium/phosphate equilibrium and provokes profound consequences on the bone metabolism, beyond having a pivotal effect as an anti-inflammatory mediator. VD deficit increases the possible occurrence of osteoporosis and several other pathologies that present alterations of bone metabolisms, such as hematological malignancies, bone marrow transplantation, inflammatory bowel diseases, and endocrinological diseases [17].

Furthermore, VD modifies relevant activities of the immune system and may change the development of immune-mediated diseases, including allergies and autoimmune diseases. VD operates by steering T lymphocytes to Th2 polarization and blocking Th1 and Th17 function and growth. The stimulation of T regulatory cells (Treg) might be its principal immunological function [18]. Numerous reports indicate a VD favorable action on diseases correlated to hyperstimulation of Th1 cells, such as psoriasis, multiple sclerosis, rheumatoid arthritis, and type 1 diabetes [19]. In allergic pathologies, where Th2 cells have a central effect, VD has a more complex effect. However, several experimentations displayed a positive action on the progression of allergic conditions, although the causal mechanisms have not always been fully explained [20].

## 3. Vitamin D and Allergies

Epidemiological data have also indicated the existence of a correlation of VD reduction with several conditions including allergies. The relationship of VD deficit and asthma is described, and several studies stated a connection with disease course worsening and a worse outcome. It is possible that VD, by improving the inflammation state, can decrease the rate of respiratory sepsis and exacerbations [21,22]. Although the correlation between VD concentrations and allergies has been questioned in a cross-sectional analysis including only allergic subjects [23], an effect of VD in the eosinophil activities is demonstrated. VD decreases the immunoglobulin E (IgE) synthesis and increases expression of interleukin-10 [24].

Other studies confirmed the correlation of VD with allergic diseases [25,26]. Some experimentations were conducted to evaluate the effect of VD in chronic urticaria (CU), allergic contact dermatitis (ACD), and atopic dermatitis (AD) [27,28], and a relationship between VD deficit with ACD and AD severity was reported [29,30], while a work proved a relevant difference between CSU subjects and controls in blood concentrations of VD [31].

An interesting analysis has reported that the period of birth and UBV exposure is correlated to the incidence of food allergy (FA). The concentrations of VD generated from skin after UBV might justify this correlation, while maternal or VD deficiency occurring in the first years of life also remarkably predisposed the FA onset in an experimental animal model [32,33].

Still in the framework of FA, employing results obtained from a nutritional investigation, it was stated that allergic sensitization to several different allergens, including foods, was more frequent in young subjects with a 25(OH)D deficit, while no relevant correlation was found between VD concentrations and FA in adults [34]. Although, a different study performed by Beak et al. reported that reduced concentrations of VD might be correlated to polysensitization of nutrition allergens [35]. Therefore, the correlation between VD and FA is still debatable.

However, other findings support the existence of a tight relationship between VD and allergies, and the beneficial action of VD supplementation in CSU subjects has been proved in several small clinical experimentations [36,37]. In a prospective research, patients suffering from CSU were treated with low or high vitamin D3 supplement for 12 weeks. All patients demonstrated a significantly decreased urticaria severity score, but the reduction was greater in patients treated with a higher VD dosage at week six. The medication score was also remarkably decreased, without distinguishing the two groups [38].

We report in the table some of the works that confirm the existence of a relationship between VD levels and allergic diseases [27,34,37,38,39,40,41,42,43,44,45,46], (Table 1).

## 4. Vitamin D and Immune Response

Several other relevant connections have been reported between blood cells, both lymphoid and myeloid cells, and VD concentrations. For instance, B cells can produce VD [47], while naïve T cells grown with VD-primed B cells demonstrated decreased proliferation, provoked by the presence of CD86 on B cells [48].

As reported above, VD has a main effect in IgE provoked responses, and it was displayed that IgE serum concentrations are enhanced in VDR-knockout animals [49]. Similarly, VD reduced IgE generation by B lymphocytes, and the IgE reaction in a type 1 allergy animal model can be altered by employing a VDR agonist [50].

Furthermore, VD can act on other effectors of the immune system, and Szeles et al. have stated that an increase of 25(OH)D provokes dendritic cells (DCs) to switch on VD related genes, and stimulation of VDRs by VD resets the DCs to turn out to be tolerogenic [51]. Remarkably, when VD was added to monocyte, DCs were less mature, presenting different amounts of MHC class II molecules and transforming CD4+ T cells to IL-10-producing Tregs [52]. Thus, VD not only produces a repressive consequence on DC development but also instructs the DCs to stimulate Tregs to generate IL-10 [53]. Almerighi et al. have confirmed that VD reduces inflammation caused by CD40L and increases IL-10 generation by CD4+ T cells [54], stimulating the expansion of Tregs presenting forkhead box P3 and cytotoxic T-lymphocyte-associated protein 4 [55] (Figure 2).

## 5. Vitamin D and Mast Cells: Effects on Allergies

VD also seems to have a role in allergic diseases and bone pathologies mediated by Th2 cells, via the IL-31/IL-33 axis [56], which is another new area to investigate the intricate overlapping mechanisms that connect allergies and osteoporosis. IL-31 is a cytokine that is able to provoke inflammation, which was proposed as a marker of tissue remodeling and different allergic and immunologic pathologies [56]. It is released by Th2 cells and, in smaller amounts, also by DCs and MCs. IL-31 stimulates eosinophils and fibroblasts, and its receptor is present in skin and endothelium. It is correlated with the onset of itch and chronic skin inflammation. An increased amount of IL-31 was reported in the skin and serum of patients suffering from AD, CSU, contact dermatitis, prurigo nodularis, cutaneous lymphoma, and mastocytosis, correlating with disease severity [57,58]. It is therefore interesting that increased concentrations of serum IL-31 are present in postmenopausal women with a reduction of bone mineral density (BMD), although there is no correlation with the degree of osteoporosis [59].

Mast cells are myeloid cells that transfer into practically all tissues, where they perform tissue–specialized evolution. They are a constituent of connective tissue and are numerous in areas such as the gut, mucous membranes, and skin [60,61], near to vessels and nerves [62]. This tactical settlement and the mediators generated from MCs explain their ability to quickly modify their milieu, and to control other forms of inflammatory cells [63,64].

MCs have a relevant effect in immediate hypersensitivity reactions, but also in late phase responses and innate immune response, by generating a large number of different mediators either from storage places in their granules or by delivering substances which are able to regulate different signaling pathways after adequate stimulation [65].

In allergic reactions, IgE joins to the IgE receptor to produce complexes on the cellular membrane of MCs to induce MC sensitization. After new contact with particular antigens, they combine with the IgE/FceRI complex to stimulate MCs [66]. Moreover, MCs also can be stimulated by temperature modifications and microbial factors [67,68].

However, beyond allergic diseases, MCs are involved in the genesis of a massive variety of pathological conditions including chronic inflammation, autoimmunity, and cancer [69].

As for the relationship between MCs and VD, the cells discharged several mediators in a VD-lacking milieu without the need of any activators. Presence of calcitriol in the culture medium increased the number of VDRs in the MCs, and VDRs provoked complexes with Lyn in MCs to block the connection of Lyn to MyD88 and to the β chain of FcεRI, which diminished the amounts of NF-kB and MAPK and reduced the phosphorylation of Syk. Moreover, VDRs connected to the promoter of TNF-α reduce the acetylation of RNA polymerase II and histone H3/H4, reducing the production of TNF-α in MCs. These findings make evident that VD is necessary to preserve the steadiness of MCs, whereas the deficit of VD provokes the stimulation of MCs [70,71].

This has been demonstrated under numerous experimental conditions. It is notorious that the release of granules and the discharge of histamine from MCs are involved in the genesis of urticaria [72,73]. VD has been suggested for this therapy and for the one concerning other allergic pathology [74,75], as MCs own the VDR capable of blocking degranulation of compounds provoked by IgE [76].

Finally, it may be important to consider how the strong correlations between MCs and VD can aid to clarify some contradictory actions provoked by MCs. Although MCs were once believed to operate essentially as cells which were able to induce inflammation that can provoke allergic responses and inflammation induced by exogenic factors such as UV irradiation [63,77], novel findings suggest that, in specific conditions, MCs can reduce cellular damage and decrease inflammation provoked by UVB irradiation [78]. Even though most UVB rays do not enter the dermis, the UVB–supported stimulation of dermal MCs is believed to be obtained through a nerve growth factor, that originated from the epidermis [79], or by nervous sensory C fibers that have been stimulated by cis-urocanic acid [80]. This secondary mechanism of MC stimulation seems to participate in the UVB-provoked systemic immunosuppression in animals exposed to an acute single dose of UVB irradiation [79]. The intimate mechanism of this condition could be constituted by the production of specific cytokines, including IL-10 (Figure 3).

MC’s generation of IL-10 can reduce the skin damage provoked by chronic UVB irradiation. Even if the specific mechanism that induces MC IL-10 generation in this condition is undetermined, it is well-known that exposure of the skin to UVB fosters the generation of 1 alpha,25[OH]2D3. A study displayed that VD could increase IL-10 mRNA generation and stimulate IL-10 delivery in animal MCs in vitro. A different experimentation evaluated the effect of VD and MCs VDR expression in UVB irradiated skin in vivo. Researchers engrafted the skin of genetically mast cell-deficient WBB6F(1)-Kit(W/W-v) mice with BM-originating MCs isolated from C57BL/6 wild-type or VDR-/- animals. MC-supported reduction of the inflammation correlated with chronic UVB irradiation of the skin in the KitW/W-v animal’s expression of VDRs by the adoptively transferred MCs. As VD can produce IL-10, these results indicate that VD/VDR-dependent stimulation of IL-10 generation by skin MCs can participate to the MCs capability to reduce skin inflammation after chronic UVB irradiation [71].

These findings suggest that stimulating the anti-inflammatory effects of MCs by adding VD might be a new strategy for decreasing tissue injury and inflammation in several different pathological conditions.

However, the relationship between mast cells and VD could be even more relevant. Yip et al. demonstrated that MCs could transform 25OHD3 to 1α, 25(OH)2D3 through 25-hydroxyvitamin D-1α–hydroxylase, and that these VD3 derivatives reduced the MCs production of pro-inflammatory mediators in a VDR-dependent modality [81]. Furthermore, when VD3 metabolites were epicutaneously employed, a remarkable decrease in skin inflammation occurred. This effect requires the presence of MCs-VDRs and MCs-CYP27B1. These results provide a mechanistic justification for the anti-inflammatory action of VD on MCs’ activity by showing that MCs can metabolize 25OHD3 to inhibit IgE-mediated MCs stimulation in both in vivo and in vitro models.

## 6. Mast Cells, Vitamin D, and Bone Metabolism

The relationship between MCs and VD goes far beyond the possibility of VD to influence the activity of MCs in allergic manifestations and both have been shown to have a role in determining bone pathology. In fact, MCs generate mediators that alter bone metabolism.

It has been known for a long time that MCs may have a role in fracture repairing [82]. Histology analysis of fractures in animal models demonstrated that, in the first fifteen days, MCs are present in the surrounding area of blood vessels and in the vascularized areas growing into subperiosteal callus [83]. This finding suggests that MCs might be implicated in the angiogenic process and in the metabolism of extracellular matrix in the initial phases of fracture repairing.

However, it was reported that MCs might reduce bone alteration repairing by activating arteriogenesis, and that the successful effect of recombinant parathyroid hormone treatment on bone healing is due to the reduction of arteriogenesis secondary to MCs inhibition [84].

Some of the effects exerted by MCs on bone metabolism may be due to the action of VD. It has been reported that MCs granules include VD, which is not produced inside the granules. VD produced by the keratinocyte transits into the intercellular milieu, and travels toward the basement membrane; at this level VD is absorbed by MCs, where it is deposited within its granules [85,86].

However, the mediators contained or produced by MCs are several hundred, and the overall effect of these cells on bone metabolism can be deeply different depending on the mediator considered and the different experimental conditions.

## 7. Mastocytosis, Allergy and Bone Alterations

The role played by MCs in allergology and on bone metabolism and any correlations with VD are highlighted in what constitutes the most serious alteration of MCs, mastocytosis, a hemopoietic malignancy characterized by the clonal proliferation of these cells and their accrual in different organs and tissues. In more than 80% of subjects with systemic mastocytosis (SM), a somatic point genetic mutation in KIT at codon 816 is discovered [87,88].

Diagnosis is performed by employing the major criterion of multifocal clusters of altered MCs in BM and the minor criteria of the occurrence of the KIT D816V mutation, increased levels of serum tryptase, and altered MC CD25 expression [89]. Cutaneous mastocytosis is most common in infancy, while systemic mastocytosis concerns adult patients and extracutaneous tissues, with or without skin participation.

The WHO classified SM into: indolent SM (ISM); smouldering SM (SSM); SM with an associated hematologic neoplasm (SM-AHN); aggressive SM (ASM); and MC leukemia (MCL). Moreover, MC sarcoma (MCS) is an extremely severe form of mastocytosis identified by a sarcoma-like proliferation of abnormal, clonal MCs, without a systemic diffusion. The outcome in these subjects is poor, and patients evolve to MCL within a brief period [90,91].

ISM is the most common form of the disease, and its symptomatology is essentially due to the release of MC mediators, discharge, and the onset of allergic manifestations. As far as the bone alterations, ISM has been identified as a risk element for reduced BMD. Considering the conventional WHO criteria [92], osteoporosis in ISM subjects goes from 18 to 28% [93,94]. According to the more recent parameters suggested by the International Society for Clinical Bone Densitometry [95], mastocytosis-correlated reduction of BMD was recognized in about 9% of women and in 28% of men with the disease [96].

However, in all subjects suffering from mastocytosis, irrespective of sex, age, or disease form, an osteopathy can occur, and VD deficit [97], osteopenia, and osteoporosis are commonly found in SM subjects [98,99,100].

In a group of patients suffering from ISM without osteoporosis, Artuso et al. studied BMD change at the lumbar spine and proximal hip before and after VD supplementation [101]. Furthermore, they investigated the possible relationships between BMD changes and serum tryptase concentrations. The presence of a VD deficit was reported in more than 70% of ISM subjects. After 30 months with VD (or calcium supplement when necessary), they reported a 2.1% increase in BMD at the lumbar spine, while no variations were detected at hip. However, after 30 months, about 60% of subjects presented VD concentrations that were still lower than recommended [101].

Other experimental data collected on animal models seem to confirm the connection between mast cell and VD during mastocytosis. Cutaneous mast cell tumors (MCT) arise more commonly in dogs than in other animals, and reduced concentrations of VD might be a risk element for MCT [102,103]. Calcitriol is recognized to possess anti-proliferative actions on several canine neoplastic cells, including the C2 cell line originated from a cutaneous MCT [104]. Remarkably, VDR is extensively present in both inflammatory and clonal canine MCs, but the recurrence rate of VDR expression is higher in tumoral MCs [105].

It is likely that the bone changes present in patients with mastocytosis have a multifactorial origin.

First, ISM-correlated osteoporosis is sustained by an increased osteoclastic activity. This supposition is also supported by the fact that anti-resorption substances are efficacious in drugs and effective in managing such conditions [94]. A recent study displayed a relationship between tryptase, osteoprotegerin, and C-terminal telopeptide of type 1 collagen, in subjects suffering from severe SM variants [26,106]. Furthermore, RANK–RANKL signaling leads the osteoclastic activity and MCs express RANK ligand, activating the osteoclasts, causing bone resorption, and osteoporosis [107]. The increase in osteoclast activity by the RANK ligand enhancement is also able to provoke the osteolytic alterations [108]. This mechanism is due to the compounds delivered by MCs. Different cytokines, such as IL-1, IL-6, and TNF-alpha, as well as tryptase, histamine, heparin, all are discharged by MCs, with direct consequences on osteoclasts and osteoblasts [100]. Indeed, these substances stimulate osteoclast function provoking bone loss and inducing osteoporosis and osteopenia, while histamine has been reported to increase bone growth causing osteosclerosis [109] (Figure 4).

Moreover, histamine working via many different receptors is answerable for the allergic effects such as pruritus, and the consequences on bronchial and gastrointestinal smooth muscle cells. Therefore, bone changes are the overall product of the chymotryptic proteases, cytokines, RANKL, fibroblast growth factor family proteins, hormonal factors, and histamine [96,110,111,112].

Another mechanism that appears to correlate mast cells, VD, allergy, and bone disease could be represented by their effects on the vascular endothelial growth factor (VEGF). VEGF is a substance able to modify vascular permeability increasing the generation of nitric oxide. Several findings demonstrated that VEGF has a fundamental action in allergic pathologies including CSU and is a blood marker for CSU identification [113]. MCs are one of the main producers of VEGF. Although VEGF is essential in bone repair and remodeling, VEGF operates as a powerful chemoattractant for osteoclasts. Moreover, VEGF production is enhanced during osteoclast differentiation via NF-κB stimulation of HIF-1α. Furthermore, osteoclasts express VEGF receptors and VEGF directly increases osteoclast survival and osteoclastic bone resorption. An experimentation stated that VD binding protein (VDBP) was increased in CSU subjects with respect to normal controls. Furthermore, sera of CSU subjects stimulated the generation of VEGF in MCs via PI3K/Akt/p38 MAPK/HIF-1α axis in an IgE-supported modality, and VD reduced the production of VEGF by blocking this axis [114]. Together, these findings indicate a possible beneficial effect exerted by the administration of VD in reducing the onset of osteoporosis in allergic patients.

In conclusion, when MCs discharge their mediators, several metabolic alterations may occur, among which are bone remodeling and osteolysis. This is a certain fact in the case of myeloproliferative neoplasms such as mastocytosis but must be further evaluated in allergic conditions. Bone remodeling is modified by MCS mediators that alter the equilibrium between bone destroying and bone generation. Mastocytosis is accompanying with great rate of recurrence to an abnormal bone mineralization [115], but bone modifications are also described in asthma and other allergic diseases. However, there is not enough data about a conclusive responsibility of MCs in osteoporosis in allergic pathologies, yet. MCs are too numerous in mastocytosis and too functioning in allergic diseases, and this indicates that the effect on bone metabolism might be analogous.

Bone alterations correlated to mastocytosis, and allergies should be managed with adequate VD and calcium supplement. Other studies are needed to evaluate the possibility of treating the bone changes in these patients, using bisphosphonates or denosumab [116].

## 8. Mast Cells, Vitamin D and Skin Diseases

The existence of MCs is a characteristic feature of different skin pathologies and tumors, and MCs can be involved in the onset and progression of tumors via the effects exerted on mitogenesis and angiogenesis, the effects on the extracellular matrix, and the effects on the immune response [117,118]. It has been theorized that MCs may have a different phenotype in several pathologic conditions and may present an immunosuppressive set-up in skin tumors, but a proinflammatory set-up in chronic skin inflammation [110]. However, the reason and the mechanisms able to cause a different phenotype are not well clarified. Exposure to UV can induce an immunosuppressive set-up in the skin, and VD might be involved in this switch [119]. In fact, as reported above, an interesting novel mechanism has been reported in animal models, in which UVB caused the generation of vitamin D3, provoking the expression of VDR in MCs and synthesis of the immunosuppressive IL-10 [120]. The leukemic HMC-1 MC line has been reported to present VDR mRNA, and isolated skin MCs have also been described to present VDR mRNA, proposing that this process could also be important in the skin [121]. However, a different mechanism may imply the MCs synthesis of vitamin D3 derivative end products, as MCs present CYP27A1 and CYP27B1 in several skin diseases [122], and these hydrolases may stimulate the synthesis of calcidiol from vitamin D3 and calcitriol from calcidiol [123]. These vitamin D3 derivatives can work as a VDR agonist ligand [124]. CYP27A1 was reported to be notably expressed in MCs BCC and SCC [122] (Figure 5).

However, the situation is probably much more complex. In fact, VDR stimulation provokes the production of CYP24A1, a hydroxylase that can block vitamin D3 derivatives. To evaluate immunoreactivity to VDR and CYP24A1 in MCs, skin biopsies were collected from the non-lesional and lesional skin of patients with BCC, SCC, actinic keratosis (AK), and psoriasis. In non-lesional skin biopsies, only 0.5–2.9% of the MCs presented CYP24A1, however, the rate of MCs presenting CYP24A1 was remarkably increased in lesional skin of SCC, BCC, and AK. In contrast to human skin, LAD2 MCs isolated from a subject with mast cell sarcoma/leukaemia demonstrated that around 34% and 6.5% of the cells were, in turn, positive for VDR and CYP24A1. Thus, while a very slight percentage of MCs in human skin present VDR and CYP24A1, the amount of MCs presenting CYP24A1 in keratinocyte skin tumors is increased [125]. The stimulation of VDR by calcitriol induces an increase of CYP24A1, in turn supposed to cause a negative regulation to provoke a reduction of calcitriol effect.

## 9. Conclusions

Experimental and epidemiological results indicate that VD deficits are correlated with the genesis of allergic diseases. In fact, VD changes immune response, MCS proliferation, and function. It also seems certain, from the data presented so far, that VD and MCs are involved in the control of bone metabolism and can intervene in the determinism of bone pathologies.

However, we are far from understanding the real mechanisms that regulate the complex interactions between MCs and VD, and sometimes contradictory data are present in the literature.

For instance, not only a VD deficiency, but also increased VD concentrations have been correlated to an increase in IgE production [126,127]. Cairncross et al. stated that increased 25(OH)D levels were correlated with a greater risk of FA in children [128], and a different study demonstrated that CD1 WT mice treated with 1α,25(OH)2D3 presented an inflammation of the skin similar to atopic dermatitis. A high dosage of 1α,25(OH)2D3 remarkably increased thymic stromal lymphopoietin mRNA concentrations. However, the effects were different in a dissimilar experimental animal model. This difference might be attributed to the heterogeneous genetic conditions of the animals [129]. Thus, both deficient and excessive amounts of vitamin D are potential risk factors associated with allergic disease.

Therefore, it is likely that different climate, seasonal, and genetic elements can affect the evaluation of the correlations occurring between MCs and VD and the consequences of their reciprocal effects on allergic diseases and bone metabolism pathologies. Further experiments will make it possible to definitively clarify these relationships and to use the administration of VD analogues for more adequate treatment of these pathologies and mast cell activation states.

## Figures and Tables

**Figure 1 biomedicines-10-01877-f001:**
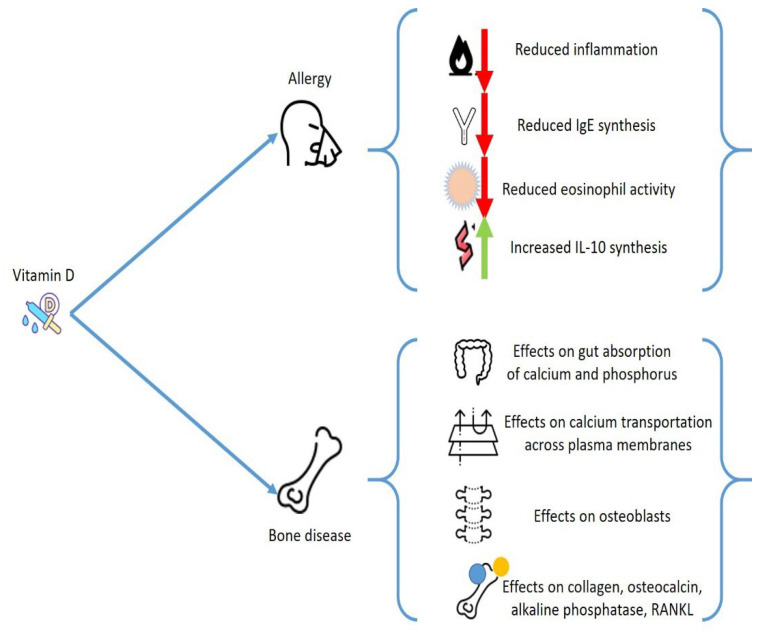
Possible effects of vitamin D on the onset of allergic diseases and on bone metabolism.

**Figure 2 biomedicines-10-01877-f002:**
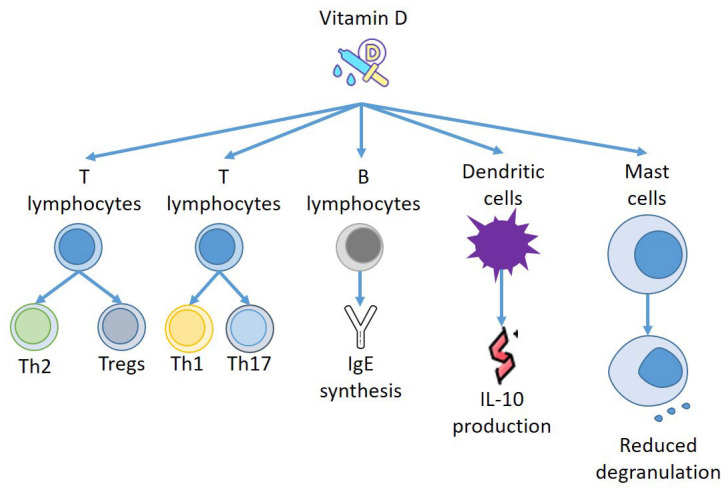
Effects of Vitamin D on immune effectors.

**Figure 3 biomedicines-10-01877-f003:**
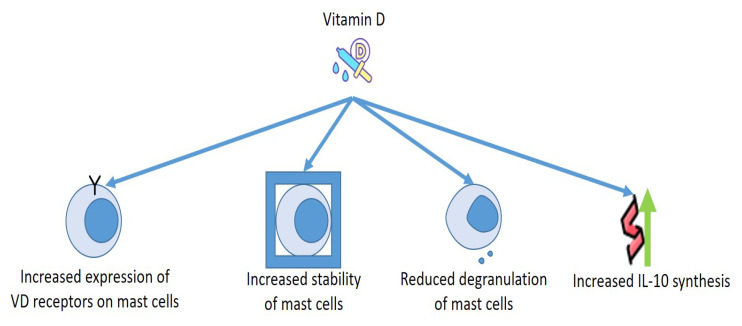
Effects of Vitamin D on mast cells.

**Figure 4 biomedicines-10-01877-f004:**
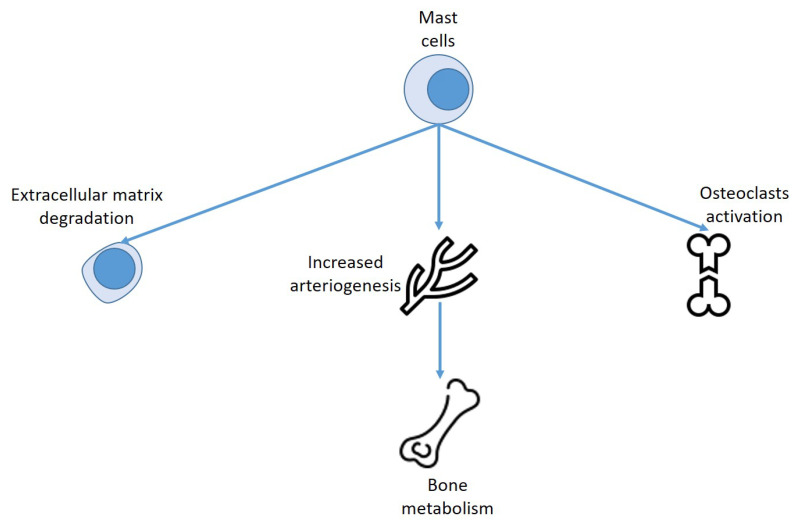
Effect of mast cells on bone metabolism.

**Figure 5 biomedicines-10-01877-f005:**
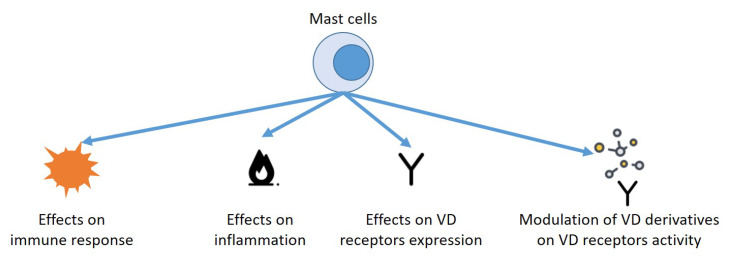
Effects of mast cells on skin disease.

**Table 1 biomedicines-10-01877-t001:** Effects of Vitamin D on allergies.

Allergic Disease	Correlation with Vitamin D	Ref.

Atopic disease	Higher values of 25OH-D3 in patients with mild disease compared with patients with moderate or severe disease	[27]

	Reduced VD levels in Atopic Disease patients	[39]

	Vitamin D supplementation during winter months had favorable effects on Atopic Disease symptoms	[40]

Food Allergy	Presence of 25OH-D3 serum values of less than 15 ng/mL	[34]

Chronic Urticaria	Reduced Urticaria Symptoms Severity scores after D3 supplementation	[37]

Chronic Spontaneous urticaria	Reduced total urticaria score after vitamin D3 administration	[38]

Asthma	Presence of 25OH-D3 serum values of less than 20 ng/mL (3.4%), and of 20–30 ng/mL (24.6%)	[41]

	Inverse correlation between maternal 25OH-D3 values and inhaled steroids	[42]

	Vitamin D supplementation during winter months reduced the frequency of asthma attacks	[43,44]

Childhood wheezing	Correlation between high maternal 25OH-D3 values with reduced childhood wheezing	[45,46]


## Data Availability

Not applicable.

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
