# Peer review of "Mast Cells and Vitamin D Status: A Clinical and Biological Link in the Onset of Allergy and Bone Diseases"

_biomedicines, 2022, doi:10.3390/biomedicines10081877_

Round 1

Reviewer 1 Report

Mast cells are important sensor and effector cells of the immune system. Increasing evidence suggests that they also play an important role in bone metabolism and bone disorders. Vitamin D is required to maintain the stability of mast cells, and Vitamin D deficiency results in mast cell activation. Vitamin D deficiency is prevalent worldwide and may partly explain the increases in asthma and allergic diseases. This review describes the connections between mast cells and vitamin D, and their activation of different molecular or cellular activation pathways, to the determination of bone pathologies and the onset of allergic diseases.

I have some suggestions:

1. The title of each section is inappropriate and does not summarize the main conclusions of the section.

 2. I suggest that the introduction section includes a diagram to summarize the overview of the relationship between vitamin D, bone pathologies and the onset of allergic.

 3. For the section 2. There are three keywords, Vitamin D, bone disease and allergy. However, there are only some information about vitamin D on immune effectors on Figure1. Please also add the information about bone disease on the Figue1.

 4. Please also provide summary figures for section 3 and section 5.

Author Response

Reviewer 1

Mast cells are important sensor and effector cells of the immune system. Increasing evidence suggests that they also play an important role in bone metabolism and bone disorders. Vitamin D is required to maintain the stability of mast cells, and Vitamin D deficiency results in mast cell activation. Vitamin D deficiency is prevalent worldwide and may partly explain the increases in asthma and allergic diseases. This review describes the connections between mast cells and vitamin D, and their activation of different molecular or cellular activation pathways, to the determination of bone pathologies and the onset of allergic diseases.

I have some suggestions:

  1. The title of each section is inappropriate and does not summarize the main conclusions of the section.

 We have changed the titles of the sections trying to make them more appropriate to the content.

For instance:

2.0 General considerations on Vitamin D

3.0 Vitamin D and allergies

4.0 Vitamin D and immune response

5.0 Vitamin D and mast cells: effects on allergies

6.0 Mast cells, vitamin D and bone metabolism

  1. I suggest that the introduction section includes a diagram to summarize the overview of the relationship between vitamin D, bone pathologies and the onset of allergic.

We have included a figure that illustrates the relationships between vitamin D, allergies and bone disease.

  1. For the section 2. There are three keywords, Vitamin D, bone disease and allergy. However, there are only some information about vitamin D on immune effectors on Figure1. Please also add the information about bone disease on the Figure 1.

 We have changed the title of the section and the effects of Vitamin D on bone pathologies have been reported in a different figure.

  1. Please also provide summary figures for section 3 and section 5.

We have added a figure for section 3 (now section 5) and for section 5 (now section 8).

Reviewer 2 Report

The authors present a review article which aims to detect the close connections between mast cells and vitamin D(VD), which contribute, through the activation of different molecular or cellular activation pathways, to the determination of bone pathologies and the onset of allergic diseases.

 Comments:

Lines 110-111,

Epidemiological data have also indicated the existence of a correlation of VD reduction with several conditions including allergies.

Lines 407-409,

6. Conclusions

Experimental and epidemiological results indicate that VD deficits are correlated to the genesis of allergic diseases.

The information of epidemiological data (effect sizes, vitamin D(VD) and allergic diseases) should be added in this review article, e.g., summary Table or Figure, by adding readability.

Author Response

Reviewer 2

The authors present a review article which aims to detect the close connections between mast cells and vitamin D(VD), which contribute, through the activation of different molecular or cellular activation pathways, to the determination of bone pathologies and the onset of allergic diseases.

 Comments:

Lines 110-111,

Epidemiological data have also indicated the existence of a correlation of VD reduction with several conditions including allergies.

Lines 407-409,

  1. Conclusions

Experimental and epidemiological results indicate that VD deficits are correlated to the genesis of allergic diseases.

The information of epidemiological data (effect sizes, vitamin D(VD) and allergic diseases) should be added in this review article, e.g., summary Table or Figure, by adding readability.

We thank the reviewer for the valuable suggestion. We have added numerous bibliographic entries and a table that makes the work more readable.

Round 2

Reviewer 1 Report

The authors answered my questions, it is recommended to accept.

Reviewer 2 Report

No further comment